Extreme dispersal or human-transport? The enigmatic case of an extralimital freshwater occurrence of a Southern elephant seal from Indiana

Valenzuela-Toro Ana M. anmavale@ucsc.edu 1 2
Zicos Maria H. 3 4
Pyenson Nicholas D. 2 5
1 Department of Ecology and Evolutionary Biology, University of California , Santa Cruz , CA , United States of America
2 Department of Paleobiology, National Museum of Natural History, Smithsonian Institution , Washington , DC , United States of America
3 School of Biological and Chemical Sciences, Queen Mary University of London , London , United Kingdom
4 Department of Earth Sciences, Natural History Museum , London , United Kingdom
5 Department of Paleontology and Geology, Burke Museum of Natural History and Culture , Seattle , WA , United States of America
Wedel Mathew
Electronic publication date: 2020 Sep 2
Publication date: 2020
Volume: 8
Electronic Location ID: e9665
Received 2020 Apr 13; Accepted 2020 Jul 15
Copyright year: 2020
Copyright holder: Valenzuela-Toro et al.
License: This is an open access article, free of all copyright, made available under the Creative Commons Public Domain Dedication. This work may be freely reproduced, distributed, transmitted, modified, built upon, or otherwise used by anyone for any lawful purpose.
License URL: https://creativecommons.org/publicdomain/zero/1.0/

Keywords: Elephant seals, Marine mammals, Historical record, Biogeography, Zoogeography, Historical ecology, Zooarchaeology, Pinnipeds, Holocene

Funding: ANID PCHA/Becas Chile, Doctoral Fellowship 2016-72170286 UK Natural Environment Research Council through the London NERC Doctoral Training Partnership NE/L002485/1 NMNH Remington Kellogg Fund Ana M. Valenzuela-Toro was funded by ANID PCHA/Becas Chile, Doctoral Fellowship; Grant No. 2016-72170286. Maria H. Zicos was funded by the UK Natural Environment Research Council through the London NERC Doctoral Training Partnership; Grant No. NE/L002485/1. Nicholas D. Pyenson was supported by the NMNH Remington Kellogg Fund. The funders had no role in study design, data collection and analysis, decision to publish, or preparation of the manuscript.

==============================
Elephant seals (Mirounga spp.) are the largest living pinnipeds, and the spatial scales of their ecology, with dives over 1 km in depth and foraging trips over 10,000 km long, are unrivalled by their near relatives. Here we report the discovery of an incomplete Holocene age Southern elephant seal (M. leonina) rostrum from Indiana, USA. The surviving material are two casts of the original specimen, which was collected in a construction excavation close to the Wabash River near Lafayette, Indiana. The original specimen was mostly destroyed for radiometric dating analyses in the 1970s, which resulted in an age of 1,260 ± 90 years before the present. The existence of sediments in the original specimen suggests some type of post depositional fluvial transportation. The prevalent evidence suggests that this male Southern elephant seal crossed the equator and the Gulf of Mexico, and then entered the Mississippi River system, stranding far upriver in Indiana or adjacent areas, similar to other reported examples of lost marine mammals in freshwater systems. Based on potential cut marks, we cannot exclude human-mediated transportation or scavenging by Indigenous peoples as a contributing factor of this occurrence. The material reported here represents by far the northernmost occurrence of a Southern elephant seal in the Northern Hemisphere ever recorded. The unusual occurrence of a top marine predator >1,000 km from the closest marine effluent as a potential extreme case of dispersal emphasizes how marine invasions of freshwater systems have happened frequently through historical (and likely geological) time.

Introduction

Elephant seals (Mirounga spp. Gray, 1827) are the largest living pinnipeds, and the spatial scale of their ecology is equally unrivalled among marine mammals: they routinely dive to over 1 km in depth—occasionally over 2 km in depth—and their foraging trips may reach more than 10,000 km (Le Boeuf et al., 2000; Robinson et al., 2012; Hindell et al., 2016). Both species in the genus Mirounga, the Northern elephant seals (Mirounga angustirostris Gill, 1866) and the Southern elephant seals (Mirounga leonina Linnaeus, 1758), show a broadly anti-tropical distribution in the Northern and Southern hemispheres, respectively (Hindell, 2018; Davies, 1958). The specific localities of rookeries for each species have shifted since they were first studied, which has prompted the suggestion that their current geographic range is largely a result of centuries or millennia of human hunting that extirpated previously occupied ranges (Rick et al., 2011). Regardless, Northern elephant seals have largely recovered from near extinction in the late 19th century, with breeding colonies in California and Mexico. Nonetheless, it is unclear whether the same is true for Southern elephant seals, whose colonies are located in southernmost Argentina and Chile and on island archipelagos throughout the Southern Ocean (Davies, 1958; McMahon et al., 2005).

Besides routine long-distance foraging, elephant seals have been recorded outside the geographic limits of their known foraging range (e.g., Elorriaga-Verplancken et al., 2020). Extralimital sightings include records of Northern elephant seals in Japan and Hawaii (Reeves et al., 2002); similarly, Southern elephant seals have been sighted along the coast of Brazil in the South Atlantic, the coast of Oman in the Indian Ocean, and even the Gulf of California in the Northern Hemisphere (Elorriaga-Verplancken et al., 2020; Johnson, 1990; Mayorga et al., 2017). Here we report the surprising discovery of an archaeological occurrence of Mirounga leonina represented by an incomplete rostrum that is Holocene age (1,260  ± 90 radiocarbon years before present, BP), recovered from the banks of the Wabash River near Lafayette, Indiana, USA. This find, located along a major river that drains most of Indiana’s watershed to the Ohio River and then to the Mississippi River, is also an unusual occurrence for elephant seals because it is located approximately 1,400 km inland in a freshwater river system.

Materials and Methods

(i) Institutional Abbreviations—FMNH, Geological Collections, Field Museum of Natural History, Chicago, Illinois, USA; LACM, Department of Mammalogy, Natural History Museum of Los Angeles County, Los Angeles, California, USA; USNM, Departments of Paleobiology and Vertebrate Zoology (Division of Mammals), National Museum of Natural History, Smithsonian Institution, Washington, District of Columbia, USA.

(ii) Material—USNM 375734 is a cast in the Smithsonian Institution’s Department of Paleobiology. The original specimen consisted of a nearly complete right maxilla with the canine in situ but lacking the postcanine dentition (Figs. 1A–1F). The original specimen was fortuitously found and collected during a building excavation in 1965, approximately 9 m below the surface of the riverbanks of the Wabash River near Lafayette, Indiana, USA (40°25′N, 86°54′W; Fig. 2B). Archival notes and correspondence housed at FMNH indicate that the bone of the original specimen was “not replaced, and therefore looks to be recent [sic] and not fossil” (CE Ray, WD Turnbull, pers. comm., 1966; see electronic File S1). Further, this documentation identifies that the original specimen had “some black pebbles and sand occluding one alveolus (potentially P1) and some small nutrient foramina” (page 11, electronic File S1), which are indicative of intermediate or relatively high ambient current energy, suggesting that the specimen underwent some type of post-mortem river transportation (Behrensmeyer, 1982; Brett & Baird, 1986). Unfortunately, the original specimen was almost completely destroyed for 14C dating analyses in the 1970s, and only a few milligrams of the original bone remains at the FMNH. The conventional radiocarbon age (1,260 ± 90 radiocarbon years BP; laboratory number Tx-1651, University of Texas at Austin; Valastro, Mott Davis & Varela, 1988) places the specimen as Late Holocene in age, prior to European contact and settlement of the Americas (see further discussion about radiometric measurements in electronic Text S1). Two plaster casts of the original specimen remain: PM 37625 at the FMNH and USNM 375734 at the Smithsonian Institution. Morphological descriptions herein were derived from USNM 375734. In addition, cast PM 37625 was evaluated by one of the authors (MZ) confirming that its morphology is indistinguishable from USNM 375734.

Figure 1 USNM 375734 in lateral, ventral, and dorsal views.

USNM 375734 in (A, B) lateral, (C, D) ventral, (E) dorsal view; (F) detail of the maxilla-premaxilla contact in dorsal view. Abbreviations: bz, base of the zygomatic process of the maxilla; if, infraorbital foramen; ip, infraorbital process; m, maxilla; p, palatal surface of the maxilla; om, orbital margin; mpc, maxilla-premaxilla contact.

Figure 2 Map showing the extant elephant seals distribution ranges in the Americas and adjacent areas and detail of the geographic location where USNM 375734 was found.

(A) Map showing the extant Mirounga leonina (orange) and M. angustirostris (purple) distribution ranges. Large circles represent their principal breeding colonies. Small circles represent extralimital sightings of M. leonina along the South Atlantic and Pacific oceans (see electronic Text S1). (B) Detail of the geographic location where USNM 375734 was found.

(iii) Specimens observed—Mirounga leonina (USNM 239141, 241199, 484893, 504927), and Mirounga angustirostris (USNM 21738, 21890, 38234, 260867, 265353, 267987, 14929, 15270, 20927, 21886, 21896, 200953, 219058, LACM 54394), Mirounga sp. indeterminate (USNM 55079).

Results

(i) Morphological description—USNM 375734 consists of a cast of a nearly complete right maxilla. Measurements of the specimen are provided in Table 1. The right canine is complete. The alveoli for P1–P4 and M1 are oval to sub-oval in shape, single-rooted and anteriorly oriented. The anterior portion of the maxilla is well preserved. The lateral surface of the maxilla is pocketed by tiny (<1 mm) foramina between the infraorbital foramen and the canine, which is a surface texture that resembles physically mature specimens of male elephant seals (e.g., USNM 260867, 484893). In dorsal view, and posterior to the level of P4, much of the margin of the maxilla appears to be missing and worn (potentially associated to some type of post-mortem transportation), exposing the inner pocketing of the external bony nares. In lateral view, the dorsal profile of the maxilla is steep in the posteriormost portion (∼50° at the level of the alveoli of M1 and P4), then anteriorly flatter (20°) until the position level with the posterior edge of the alveolus of P1, where the angle becomes intermediate before finally curving around the alveolus of the canine. The infraorbital foramen is dorsoventrally compressed (measuring ∼20 mm in diameter in the widest orientation and ∼15 mm in height), and it is visible in lateral, anterior and oblique ventral views. The base of the zygomatic process of the maxilla is robust and rugose in its inner surface. The preorbital (=infraorbital) process of the maxilla is conspicuous, though slightly worn, and it is located ∼2 cm dorsal of the base of the zygomatic process. Together, these two structures underlie an apparent large and elevated orbit (i.e., situated above the level of the alveoli row). In ventral view, the posterior border of the maxilla is semi-circular, and potentially constitutes the joint line between the maxilla and the palatine. USNM 375734 likely represents a physically mature individual because of its large size, the lateral surface texture, the relatively static position of the canine in its alveolus, and the possession of clear interalveolar septa between the postcanine alveoli. Furthermore, USNM 375734 likely represents a male based on qualitative and quantitative comparisons (see Fig. 3).

Table 1 Morphological measurements (in cm) of USNM 375734.

Maximum length of the maxilla	18.2	
Canine mesiodistal diameter	4.3	
Canine buccolingual diameter	3.29	
P1 alveolus mesiodistal diameter	1.41	
P1 alveolus buccolingual diameter	1.19	
P2 alveolus mesiodistal diameter	1.36	
P2 alveolus buccolingual diameter	1.02	
P3 alveolus mesiodistal diameter	1.29	
P3 alveolus buccolingual diameter	0.89	
P4 alveolus mesiodistal diameter	1.11	
P4 alveolus buccolingual diameter	0.83	
M1 alveolus mesiodistal diameter	1.18	
M1 alveolus buccolingual diameter	0.76	
Length of the tooth row	8.08	
Length of the infraorbital foramen	1.93	
Height of the infraorbital foramen	1.46	

Figure 3 Canine measurements for Northern and Southern elephant seals.

Upper canine measurements for modern Northern and Southern elephant seals and USNM 375734. Data from electronic Table S1. Squares and triangles represent Southern and Northern elephant seals, respectively. Green and gray colors represent females and males, respectively. USNM 375734 is represented by a black dot.

USNM 375734 exhibits a semi-linear striation transversally directed in the dorsolateral surface of the maxilla that is approximately 2 cm long (Fig. 4). In addition, three fine semi-parallel striae (∼2 cm long) occur in the medial section of the maxilla, which align with the major striation described above. The narrow, shallow and linear aspect, together with the parallel distribution of the striae suggest that they would correspond to intentional cuts by a sharp-edged implement (e.g., stone tool, but not a metal tool) (Fisher, 1995; Greenfield, 2006; Domínguez-Rodrigo & Baquedano, 2018). However, the loss of the original specimen prevents us from performing a more detailed examination of these striae (e.g., scanning electronic microscopy).

Figure 4 Potential cut marks in USNM 375734.

USNM 375734 in dorsomedial view (A). A magnification of the square is shown in (B). The black arrows heads indicate locations of the potential cut marks described in the text.

(ii) Remarks—USNM 375734 belongs to Phocidae, based on the anterior extension of the rostrum relative to both the (inferred) position of the nasals, and the position of the bony naris, as well as the relatively large size of the infraorbital foramen. Additionally, USNM 375734 belongs to Mirounga based on: a combination of an overall large size with an enlarged canine tooth; its toothrow anteriorly oriented; and the possession of single-rooted postcanine alveoli (King, 1972). The principal osteological differences between M. angustirostris and M. leonina relate to traits associated with the elongation of the rostrum and morphology in the ventral surface of the skull (Briggs & Morejohn, 1976). M. leonina has a foreshortened skull compared to M. angustirostris, which is reflected in differences in the relative position and elongation of different structures in the anterior section of the rostrum. For instance, the position of the last molar with respect to the level of the infraorbital foramen constitutes a diagnostic trait to differentiate between the species, with M. leonina having M1 located at the same level as the infraorbital foramen, whereas M. angustirostris has it located anteriorly to this level (Briggs & Morejohn, 1976). In this regard, USNM 375734 mirrors the morphology of M. leonina in having the alveolus of M1 aligned with the infraorbital foramen. Furthermore, the dorsal aspect of the anterior section of the maxilla, where it contacts the premaxilla, is anteroposteriorly short and has a transversely oriented posterior margin, giving it a relatively square appearance, parallel to M. leonina, and contrasting with M. angustirostris, in which this margin is oriented posterolaterally and has a more rectangular shape. The shape of the dorsal profile of the maxilla (i.e., steep in the posteriormost portion, then anteriorly flat, and then curving around the alveolus of the canine) also contrasts with adult male M. angustirostris, in which the dorsal aspect is steep but straight (e.g., USNM 219058 in the Division of Mammals, and LACM 54394). Unfortunately, other diagnostic traits could not be evaluated because of the fragmentary nature of the specimen reported here. However, we consider that the relative position of the alveolus of M1 relative to the infraorbital foramen, the orientation and shape of the contact between the premaxilla and the maxilla, and the dorsal aspect of the maxilla as the most salient comparative traits, and they support our differential diagnosis and identification of USNM 375734 as a Southern elephant seal.

(vi) Radiocarbon date—The radiocarbon measurement Tx-1651 (Valastro, Mott Davis & Varela, 1988) was collected prior to this study, in 1972 or 1973, by a laboratory at the University of Texas at Austin (FMNH archives; see electronic File S1). This laboratory used liquid scintillation counting of benzene with lithium carbide and vanadium catalysts; they obtained the conventional date of 1,260 ± 90 yrs BP from the apatite portion of the bone, reported with 1 σ. Their calculation was based on 14C half-life of 5,568 years and “modern standard of 95% NBS oxalic acid, supplemented by tree rings of pre-industrial wood from a log cut in the 1850s” (Tx-540; see Valastro & Mott Davis, 1970; Valastro, Mott Davis & Varela, 1988). Although no pre-treatment is mentioned in the date list, the laboratory followed Haynes (1968)’s acid pre-treatment of apatite in previously measured bones (Valastro & Mott Davis, 1970); we infer that the original specimen may have been similarly pre-treated. No value for the organic or collagen portion of the bone was reported; we thus assume that these analyses were not undertaken at the time. Lastly, Valastro, Mott Davis & Varela (1988) state that no measurements of 12C or 13C were collected, and that no corrections were made for fractionation of 13C (Valastro, Mott Davis & Varela, 1988).

Discussion

Living Southern elephant seals have a nearly circumpolar distribution in the Southern Hemisphere with breeding colonies in the southernmost part of Chile and Argentina and Sub-Antarctic islands. We argue that the preponderance of evidence suggests that USNM 375734 represents an adult male Southern elephant seal, which was an errant migrant that swam northward from the South American coast into the Mississippi River system via the Gulf of Mexico, and eventually stranded upriver in Indiana where it was killed or scavenged by Indigenous peoples. We base our interpretations on the combination of biological and life history, and physical evidence as follows.

First, Southern elephant seals are capable of performing long foraging migrations that can extend several thousand of kilometres, spending, on average, almost 10 months of the year at sea (Hindell & McMahon, 2000; Hindell et al., 2016). In fact, extralimital occurrences at very distant locations from their known distribution are not uncommon (see below). Moreover, the occurrence of oceanographic events affecting the prey distribution (e.g., La Niña cold conditions) have been suggested as key factors driving these abnormal occurrences along the South Pacific and Atlantic oceans (see Elorriaga-Verplancken et al., 2020). Second, the radiometric age of the specimen is contemporaneous in time (∼1200 years BP) with native Mississippian civilizations, including the Cahokia culture that inhabited the Mississippi River region between ∼700–1,400 of the Current Era (Pauketat, 2004). While radiocarbon dates obtained from the original bone (and in particular, apatite) from the early 1970s are considered unreliable due to a combination of technological and methodological concerns (see electronic Text S1), there is no way to resample the original specimen (see below); also, we note that marine mammal material is especially susceptible to the marine reservoir effect, which biases radiocarbon dates towards older values (Stuiver & Polach, 1977). The original analysis did not correct the conventional date for the marine reservoir effect, but corrections for marine mammal material have reduced the conventional (uncalibrated) age from 200 years and younger to upwards of 800 years, depending on the local magnitude of the reservoir effect (e.g., Dyke, McNeely & Hooper, 1996; Koch et al., 2019; Olsson, 1980); by implication, the original specimen may be several hundred years younger than ∼1200 years BP. Third, the presence of potential intentional cuts marks by some type of stone tool (see Fig. 4) prevents us from excluding some degree of human-mediated transportation. Unfortunately, the absence of original skeletal material precludes more exhaustive interpretation and study of these marks. Furthermore, the complete lack of carnivore scavenging marks suggests a short post-mortem period before burial for this individual; yet, the existence of sediment (i.e., pebble and sand) occluding one alveolus indicates that the specimen underwent some type of post-depositional fluvial transportation, likely in Indiana (although we cannot exclude the possibility of elsewhere).

Another, and less likely explanation, is that the skull was collected by Indigenous peoples along the coast of South America or at an intermediate point (including the Mississippi drainage or coastal Americas), and then transported (singularly in one lifetime, or across a trade route) until its deposition on the banks of the Wabash River. In this regard, Mississippian societies were characterized by long distance trading networks of several goods, principally mineral resources like galena, copper, and salt, as well as marine shells which had high socioeconomic importance (Prentice, 1987; Trubitt, 2005). Furthermore, the archaeological record indicates that only skeletal elements of high socioeconomic value (e.g., isolated canines) would likely be transported long distances and traded by Indigenous people in the Americas (Lyman, 1994). We thus judge this potential scenario as less likely of an explanation than a basic biological one.

Although the morphological evidence strongly supports our identification of USNM 375734 as a male Southern elephant seal, it should be noted that its correspondence to a Northern elephant seal cannot be entirely excluded. Considering that their current and historical range is delimited to the eastern current systems of the North Pacific Ocean and its coastlines, the potential finding of a male Northern elephant seal in Indiana could be explained through two non-exclusive scenarios. For one, an individual could have swum northward along the western coast of North America, reaching the Arctic, and then crossing the Arctic to the Atlantic. Breeding colonies of Northern elephant seals are located along the Californian coast up to Mexico; however, male individuals perform long distance post breeding foraging migrations, reaching the coast of Alaska and the Aleutian Islands, and even of Japan and Russia (Le Boeuf et al., 2000; Fomin & Burkanov, 2019). Thus, an errant male could have transited along the northernmost border of North America through the Arctic reaching the North Atlantic Ocean, to finally enter into a river system via the Gulf of St. Lawrence or the Gulf of Mexico, where it could it be killed or scavenged by Indigenous peoples. Nevertheless, we consider this scenario unlikely because there are no records of this type of dispersal (i.e., throughout the Arctic) for elephant seals. Secondly, a male Northern elephant seal could have been killed or scavenged by Indigenous peoples somewhere on the western coast of North America and then transported and/or traded into the Mississippi River region. If this is the case, we would expect to find similar archaeological or historical records of marine mammal fauna from the North Pacific in the Mississippi Region; however, we did not find previous reports supporting this hypothesis. Furthermore, Rick et al. (2011) have suggested that during much of the Holocene, the abundance and distribution of Northern elephant seals along the western coast of North America were different than today, being characterized by a low abundance along the coast of California, reducing the likelihood of this scenario even more.

Molecular methods to confirm or refine the taxonomic identifications, such as ancient DNA or collagen fingerprinting, were not applied because the surviving material of the original specimen at FMNH is insufficient for analysis. This material is unlikely to yield results; or, at best, unlikely to yield results that would enhance our morphological assessment. First, it is unknown whether the original bone was well preserved and likely to yield DNA or collagen. Second, the remaining material consists of a few milligrams of bone chips and small gravel, probably from the external part of the bone, which is usually removed during sampling for DNA (e.g., Brace et al., 2012; Frantz et al., 2016; Rohland & Hofreiter, 2007). This material is far less than the ideal minimum of 10–20 mg of bone powder obtained from dense bone after contamination-removal methods (Dabney & Meyer, 2019), suggesting that the surviving bone fragments of the original specimen are very unlikely to yield endogenous DNA or collagen. Collagen fingerprinting, based on spectroscopy of collagen, has been conducted on both M. angustirostris and M. leonina, and their collagen spectra appear indistinguishable (Hofman et al., 2018). These latter concerns, even if sufficient material from the original specimen were available, mitigate against the utility of these techniques to advance with the taxonomic identification in this case.

Interestingly, the historical record provides many accounts for extralimital records in pinnipeds. For instance, hooded seals (Cystophora cristata (Erxleben, 1777)), native to North Atlantic and polar waters, have been recorded as far south as the Caribbean (Mignucci-Giannoni & Odell, 2001) and the Pacific Ocean (Dudley, 1992). Similarly, Antarctic seals such as Weddell seals (Leptonychotes weddellii (Lesson, 1826)) have been recorded off the coast of Brazil (Frainer, Heissler & Moreno, 2018) and New Zealand (Miskelly, 2015). Tropical species also have vagrant individuals, as it is shown by a record of a Galapagos fur seal (Arctocephalus galapagoensis Heller, 1904) in Guatemala (Quintana-Rizzo et al., 2017). In particular, vagrant Southern elephant seals have been recorded on the coasts of Argentina, Uruguay, Brazil, Chile, Peru, Ecuador, Galapagos Islands, Panama, South Africa, Australia, New Zealand, Oman, and Mexico (Fig. 2A) (Alava & Carvajal, 2005; Magalhães et al., 2003; Mayorga et al., 2017; Redwood & Felix, 2018; Acevedo et al., 2016; Johnson, 1990; Sepúlveda et al., 2007; De Moura et al., 2010; Pacheco, Silva & Riascos, 2014; Cárcamo et al., 2019; Sepúlveda et al., 2018; De Moura, Di Dario & Siciliano, 2011; Mertz & Bester, 2011; Mayorga et al., 2016; Shaughnessy, Kemper & Ling, 2012; Elorriaga-Verplancken et al., 2020).

Nevertheless, the material reported here represents by far the northernmost occurrence of a Southern elephant seal in the Northern Hemisphere, and potentially the most extreme freshwater incursion of a marine pinniped ever recorded. Notably, inland Holocene marine mammal localities have previously been reported across the USA, including Holocene and Pleistocene age manatee (Trichechus manatus Linnaeus, 1758) material from the Ohio and Mississippi rivers (Williams & Domning, 2004; Baghai-Riding et al., 2017), although these finds are consistent with routine freshwater incursions that manatees often undertake. More strikingly, Holocene age bowhead (Balaena mysticetus Linnaeus, 1758), rorqual (Balaenopteridae sensu lato) and sperm whale (Physeter macrocephalus Linnaeus, 1758) material have been reported from Michigan (Harington, 1988). Although Harington (1988) considered each occurrence to have been human-mediated transport, elsewhere in North America Pleistocene and Holocene cetacean material is not uncommon, especially, adjacent to large river drainages (Harington, 1988). Prehistoric harbor seals (Phoca vitulina Linnaeus, 1758) have been reported along the Columbia River of Washington State (USA), at a maximum of ∼300 km upriver from the closest marine effluent source (Lyman et al., 2002). However, these are shorter distances than the putative one for the Indiana specimen (∼1,400 km). While the incursions of Columbia River harbor seals were likely limited by waterfalls (Lyman et al., 2002), the incursions represented by manatee material from Ohio (Williams & Domning, 2004), and a common bottlenose dolphin from Tennessee (JG Mead, pers. comm., 2019) indicate that there is no obvious physical barrier for marine mammals to make long-distance upriver incursions in those rivers (including the Wabash River), at least in the Holocene, and likely in the Pleistocene as well. Further, the record of vagrant elephant seals from the Guayas River Estuary Basin (Gulf of Guayaquil) in Ecuador implies an extralimital dispersion over a geographic range of ∼8,000 km from the circumpolar region to tropical freshwater systems in the Gulf of Guayaquil (Páez-Rosas et al., 2018). These confirmed sightings were made upwards of 50 km in the river system, and animals were not in poor body condition (Páez-Rosas et al., 2018), meaning that Southern elephant seals can travel upriver with no serious detrimental effects to their health. This latter occurrence, and the fact that other phocids use freshwater habitats (e.g., harbor seals in Canada and Lake Saima ringed seal Pusa hispida saimensis (Nordquist, 1899)), suggests an absence of physiological restrictions for seals moving upriver as well.

Departures from routine dispersal or migratory patterns in marine mammals, including Southern elephant seals, are surprising but not uncommon (e.g., Lyman et al., 2002; Lucero et al., 2018; Geijer, Notarbartolo di Sciara & Panigada, 2016; Weller et al., 1996; Elorriaga-Verplancken et al., 2020). In this study, the occurrence of a top marine predator at more than 1,000 km from the closest marine effluent source shows that the dispersal behavior of elephant seals is more dynamic and complex than previously recognized, suggesting a reconsideration of the current assumptions of elephant seal (as well as other marine mammals) life history patterns. Many questions arise from this finding, including the evolutionary and conservation implications of these extreme dispersal behaviors: is this behavioral plasticity partially responsible for the ongoing population recovery and recolonization of elephant seals after their near extermination during the last century? Could this dispersal ability make elephant seals more resilient to oceanographic and ecological changes resulting from climate change? Ultimately, advances in telemetry techniques, stable isotope analysis, and ancient DNA will provide critical information for a better understanding of these enigmatic long dispersals of top marine predators (Harcourt et al., 2019; Pinsky et al., 2010).

Conclusions

We presented the finding of a cast of a Holocene rostrum of a Southern elephant seal (Mirounga leonina) from Indiana, USA. The radiometric dating of the original specimen prior to this study resulted in a conventional age of 1,260 ± 90 years before the present. The specimen was identified as an adult male Southern elephant seal based on the size of the canine, the relative position of the alveolus of M1 relative to the infraorbital foramen, the orientation and shape of the contact between the premaxilla and the maxilla, and the dorsal aspect and lateral surface texture of the maxilla. Considering the natural history and ecology of modern Southern elephant seals, we propose that this adult male crossed the equator and the Gulf of Mexico, and entered the Mississippi River system, stranding far upriver in Indiana or adjacent areas. Nevertheless, because of the occurrence of potential cut marks, we cannot exclude human-mediated transportation or scavenging by Indigenous peoples as a contributing factor to this occurrence. This record represents the northernmost occurrence of a Southern elephant seal ever recorded and suggest that invasions of freshwater systems by marine mammals have happened frequently through time.

Supplemental Information

File S1 Reproduction of original documents and correspondence associatedwith USNM 375734 referring to the discovery and study of the original specimen found in Indiana

Click here for additional data file.

Text S1 Supplementary discussion of the radiometric measurements of USNM 375734

Click here for additional data file.

Table S1 Comparative measurements of elephant seals

Upper canine measurements in cm for modern specimens of Northern and Southern elephant seals and USNM 375734. Data of Mirounga leonina from Koch et al. (2019). Data of Mirounga angustirostris from this study.

Click here for additional data file.

We are thankful to P Koch, D Costa, T Keates, A Favilla, T Rick, B Pobiner, and EG Veatch for comments that improved early versions of this study. Finally, we thank the editor and two anonymous reviewers whose comments significantly improved this manuscript.

Additional Information and Declarations

Competing Interests

Author Contributions

Data Availability

Nicholas D. Pyenson is an Academic Editor for PeerJ. Besides that, the authors declare that they have no competing interests.

Ana M. Valenzuela-Toro performed the experiments, analyzed the data, prepared figures and/or tables, authored or reviewed drafts of the paper, and approved the final draft.

Maria H. Zicos and Nicholas D. Pyenson conceived and designed the experiments, performed the experiments, analyzed the data, authored or reviewed drafts of the paper, and approved the final draft.

The following information was supplied regarding data availability:

The reported specimens in this manuscript are permanently deposited at the Field Museum of Natural History, Chicago, Illinois, USA, and the National Museum of Natural History, Smithsonian Institution, Washington, District of Columbia, USA. They can be accessed by their collection numbers PM 37625 and USNM 375734, respectively.

The correspondence associated with the original specimen is available in File S1. The raw comparative measurements of Southern and Northern elephant seals used for making Fig. 3 are available in Table S1.

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
