# Peer review of "Extreme dispersal or human-transport? The enigmatic case of an extralimital freshwater occurrence of a Southern elephant seal from Indiana"

_PeerJ, doi:10.7717/peerj.9665_

## Round 0.1 · original submission · Major Revisions

This is a very interesting find and I'm glad that you are working to get it published. However, I agree with the reviewers that more work is needed. One reviewer finds the human transport hypothesis interesting, and would like more information. The other finds it unsupported, and would like it to be better supported or stricken. I think it is sufficiently plausible to warrant discussion, but I agree that more work is needed, especially regarding the possible cut marks (useful references are suggested in the reviews).

Although it is not a requirement, I think you might find it valuable to explicitly frame the long-distance dispersal and human transport hypotheses, lay out the evidence for each one (I realize much of this will be circumstantial and based on other similar cases), and explicitly state what it would take to falsify each one. If you cannot falsify either one, that is not a defeat! As long as your evidence and reasoning are clear, ending with two or more possible explanations may just be good science. But as I said, this is just a suggestion, not a requirement, as long as you can satisfactorily address the reviewers' concerns, either in the revised manuscript or the rebuttal letter.

I would like to see essentially all of the supplementary material incorporated into the main manuscript, especially the figures. I can only assume they were omitted to meet a draconian length requirement at some other venue. That requirement no longer exists, and the figures are so important, particularly the photos showing the possible cut marks, that it would criminal not to include them in the main body of the paper. The only element that should probably remain in the SI is the correspondence, which is interesting and valuable.

Oh, and you have it correctly: the plural of septum is septa, not septae.

I look forward to seeing an improved version of this work in the near future.

Reviewer 1 ·

Basic reporting

The manuscript is generally well-written in clear professional English. The Ms. is well-structured with relevant tables, figures and supplemental data.

There should be more references on marine mammal morphometrics and cut mark analysis - I explain below.

The results are relevant to the hypothesis but a more rigorous morphometric analysis should be conducted.

The inclusion of the original documentation and correspondence deserves high praise. More attention could be paid to the contextual and condition discussions in the original correspondence. Context is important for this specimen, yet remains ill-defined.

Experimental design

The identification of the maxilla needs to be more rigorous. The key character used seems to be a qualitative measure of the position of M1 relative to the infraorbital foramen. This can be better demonstrated by a detailed morphometric analysis of the distance between the lowest point of the IF to the center of the labial surface of the M1 alveolus. I think more detailed analysis of other morphological characters of the maxilla are warranted as well.

The original collection context and condition of the specimen is problematic and needs to be better explained. Some critical information is available in the original correspondence concerning both. The inclusion of this information is both important, laudable and could be discussed more in the manuscript.

It remains to be determined if the putative cut marks on the maxilla are actually cut marks. If so, human agency makes the this story even more complicated. It may be difficult to rule out trade of the maxilla from the gulf coast or even the west coast as opposed to an extraordinary dispersal event.

Validity of the findings

The validity of the findings depends on the rigor of the identification and evaluation of the context of the specimen and the analysis of the putative cut marks. More detail in these areas would make the hypothesis testing more replicable.

Good morphometric analysis illustrates that the canine (and maxilla) represents a male Mirounga. I am left wondering if more detailed morphometric analysis of the maxilla might refine the identification - helping to rule out/in the potential contribution M. angustirostris as discussed in the ned of the manuscript.

Annotated reviews are not available for download in order to protect the identity of reviewers who chose to remain anonymous.

Reviewer 2 ·

Basic reporting

The manuscript by the authors is relatively well written, with sufficient citations--although some sections would benefit from the inclusion of additional sources--and includes the important figures, tables, and raw data for reproducibility, with revisions these data should be published. However, the study requires clarifications related to the data presented, especially the context and convenience of the specimen, radiocarbon dates, including calibration software, isotopic values, and taphonomic analysis of purported cut marks.

Experimental design

The study could benefit from the clarification of methods, especially the analysis of the "cut marks," which make up a significant component of the study. Also, was the second cast of the specimen analyzed? If not, the audience should know.

Validity of the findings

The findings, as currently reported, are not supported by the data presented by the authors, especially the interpretation that the elephant seal specimen was scavenged, butchered, or killed by indigenous peoples. There is no evidence to support the interpretation.

Additional comments

Thank you for the opportunity to review your manuscript.

I have made revisions to the document in the attached document. My recommendation is that the manuscript is accepted pending major revisions.

Some minor notes:
1) please revise the manuscript for grammatic issues, spelling, and sentence structure.
2) The manuscript could benefit from thorough editing to make it concise and remove redundancies and inconsistencies in reporting. These inconsistencies raise "red flags" for the reader
3) Please thoroughly revise the supplemental sections. There were inconsistencies in the manuscript. See attached.

Major issues:
1) See my notes in the attached documents for all the concerns I have with the manuscript.

Annotated reviews are not available for download in order to protect the identity of reviewers who chose to remain anonymous.

---

## Round 0.2 · accepted · Accept

Thank you for your diligence in addressing my concerns and those of the reviewers. I am satisfied with the revised manuscript, and I am happy to accept it for publication in PeerJ.

The decision of whether or not to publish the peer reviews alongside the paper is entirely yours, and will not affect how your paper is handled going forward. However, I encourage you to do so. Making the reviews public allows the reviewers to receive credit for their efforts, and also contributes to the emerging culture of fairness and transparency in editing and peer review.